# The Fuel Moisture Index Based on Understorey Hygrochron iButton Humidity and Temperature Measurements Reliably Predicts Fine Fuel Moisture Content in Tasmanian *Eucalyptus* Forests

David M. J. S. Bowman *, James M. Furlaud, Meagan Porter and Grant J. Williamson

School of Natural Sciences, University of Tasmania, Private Bag 55, Sandy Bay, Hobart, TAS 7001, Australia
* Correspondence: david.bowman@utas.edu.au; Tel.: +61-4288-94500

**Abstract:** Fine fuel moisture content (FFMC) is a key determinant of wildfire occurrence, behaviour, and pyrogeographic patterns. Accurate determination of FFMC is laborious, hence managers and ecologists have devised a range of empirical and mechanistic measures for FFMC. These FFMC measures, however, have received limited field validation against field-based gravimetric fuel moisture measurements. Using statistical modelling, we evaluate the use of the relationship between gravimetric FFMC and the Fuel Moisture Index (FMI), based on Hygrochron iButton humidity and temperature dataloggers. We do this in Tasmanian wet and dry *Eucalyptus* forests subjected to strongly contrasting disturbance histories and, hence, percentage of canopy cover. We show that 24 h average FMI based on data from Hygrochron iButtons 0.75 m above the forest floor provides reliable estimates of gravimetric litter fuel moisture (c. 1 h fuels) that are strongly correlated with near surface gravimetric fuel moisture sticks (c. 10 h fuels). We conclude FMI based on Hygrochron iButton data provides ecologists with an economic and effective method to retrospectively measure landscape patterns in fuel moisture in Tasmanian forests.

**Keywords:** iButton datalogger; fire danger index; fuel moisture stick; humidity; temperature; microclimate; meteorological data; wildfire

## 1. Introduction

Living and dead fuel moisture is a key determinant of both wildfire occurrence and behaviour [1,2]. Globally, across biomes, dead fuels below a moisture threshold of c. 10% become available to burn; hence, under drought conditions, plant communities that are typically too moist to burn, such as rainforests and swamplands, can support wildfire [3]. The moisture content of fine fuels (e.g., dead leaves, bark, twigs, and dead grass) is widely recognised as determining whether a vegetation type is flammable or not. Hence, understanding the geographic patterns and temporal trends of fine fuel moisture content (FFMC) is a prerequisite for accurate predictions of fire risk, spread, and behaviour [1,2,4,5]. Despite being a basic variable in fire ecology and management, there remains limited empirical field data on the geographic and temporal patterns of FFMC for most Australian plant communities [6–11]. An important reason for this gap in knowledge is that measurement of the moisture content of different fuel components, or fuel moisture (or hazard) sticks [12], is laborious, requiring frequent field sampling. Furthermore, determination of gravimetric FFMC has a substantial time lag given the requirement for 24 h oven drying [1,2,4,10,13].

To overcome these practical difficulties, managers and ecologists have devised various means of estimating fuel moisture based on the physical principle that fuel moisture equilibrates with microclimate humidity and temperature [1,2]. The time of equilibration is used to denote different fuel types: fine woody fuels equilibrate <10 h, and leaf litter <1 h [1,2,14]. For instance, electronic fuel moisture sticks have been shown to be an

effective tool for fire managers who require real-time measurement of 10-hour FFMC [12], albeit these sensors require calibration to suit specific vegetation types and are expensive to deploy [4].

The availability of cheap and robust temperature and humidity dataloggers, such as the Hygrochron iButton [15], now enable ecologists to make retrospective studies of the geographic and temporal variation in FFMC, using the elegantly simple Fuel Moisture Index (FMI, Equation (1)) [16].

$$\text{FMI} = 10 - 0.25\,(T - H) \tag{1}$$

where $T$ is the air temperature (°C) and $H$ is the percentage of relative humidity (RH). This index is dimensionless but can be recalibrated to be expressed as percentage of fuel moisture content, by establishing a regression relationship with a range of existing fuel moisture models and empirical data; although the scaling factors based on these relationships are specific to different regions and forest types [8,10,17].

Furlaud et al. [18,19] used FMI based on Hygrochron iButtons' humidity and temperature sensors, suspended c. 0.75 m above the ground, to estimate variation in FFMC across macroecological gradients and disturbance gradients in tall, wet *Eucalyptus* forests. Nonetheless, FMI has received limited field validation against gravimetric FFMC. While few studies have performed field validations of FMI against gravimetric FFMC, those that have found it to be among the best predictors of fuel moisture content, despite its relative simplicity [8,17]. For instance, the detailed micro meteorological study by Nyman et al. [10] found the relationship between daily mean gravimetric FFMC and FMI was variable depending on where temperature and relative humidity measurements were measured. They found the best estimate of gravimetric FFMC was FMI based on temperature and RH measurements using Hygrochron iButtons within the litter pack, housed in specially designed containers to protect them from becoming saturated during heavy rainfall events [10]. These authors also found that there were systematic differences between vegetation types.

Here, we perform a simple analysis of the relationship of measured of fuel moisture derived from gravimetric FFMC, with the Fuel Moisture Index [16] calculated from humidity and temperature measurements using Hygrochron iButton in wet and dry *Eucalyptus* forests that surround Hobart, the capital of Tasmania. The wet forests in our study were dominated by *Eucalyptus obliqua* with a *Nematolepis squamea* (wet sclerophyll) understorey. The dry forests were dominated by *Eucalyptus globulus* and *E. pulchella* with an Allocasuarina verticillata understorey. The study was conducted over the austral summer of 2021–2022. The analysis had three stages:

(1)   Correlating between gravimetric moisture content of leaf litter (c. 1 h fuels) with gravimetric fuel moisture sticks (c. 10 h fuels [12]) 30 cm above the litter layer;
(2)   Contrasting the gravimetric litter fuels (c. 1 h fuels [2]) amongst the seven communities, with contrasting vegetation structures associated with ecological and management factors (Figure 1), measured over the summer field campaign;
(3)   Comparing estimates of FFMC using Hygrochron iButtons positioned in litter pack and 0.75 m above the litter layer.

To contextualise the study period, we provide estimates of the soil dryness index (SDI) [20], which is a key input into the forest fire danger index (FFDI), based on meteorological data from the nearby Bureau of Meteorology's Hobart station (Figure 1). FFDI is a widely used Australia index of fire weather that is linked to fire behaviour in *Eucalyptus* forests, based on air temperature, humidity, wind speed, and a proxy for fuel moisture using a drought factor, such as the SDI [20], that is based on antecedent rainfall [21].

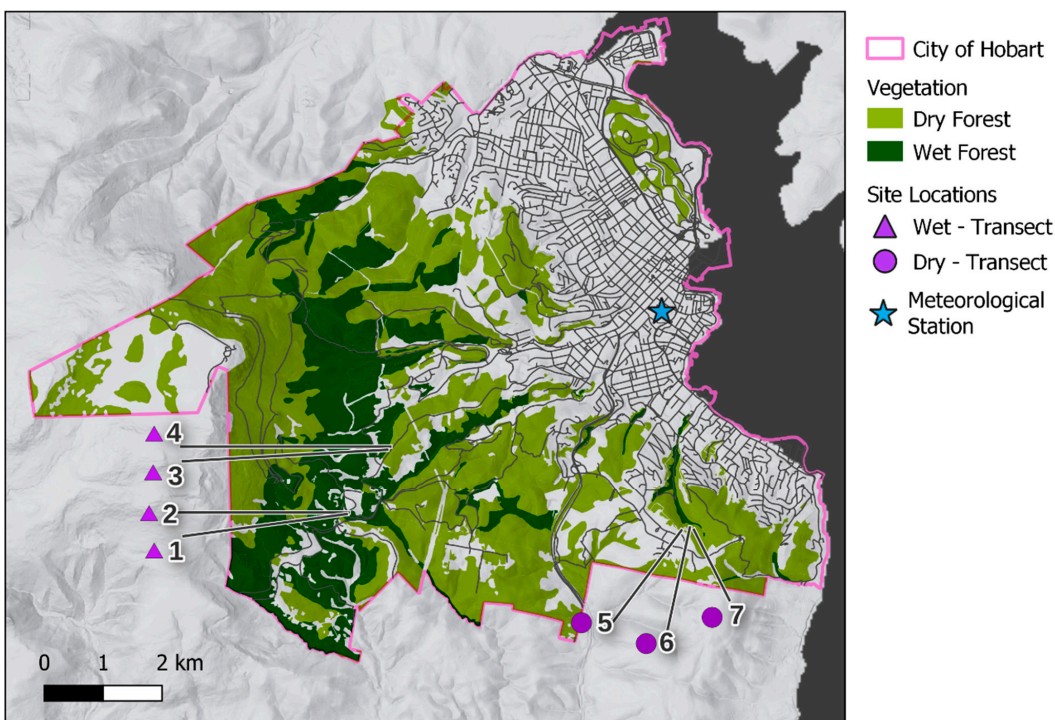

**Figure 1.** Map of city of Hobart study area with location of wet and dry forests and non-forest land use. Transect site locations (numbered 1 to 7) are indicated, refer to Figure 2 for details. The location of the meteorological station used for the soil dryness index is indicated; note it is not representative of the forest environment, being at lower altitude and in close proximity to the coastal environment that experiences a more maritime climate.

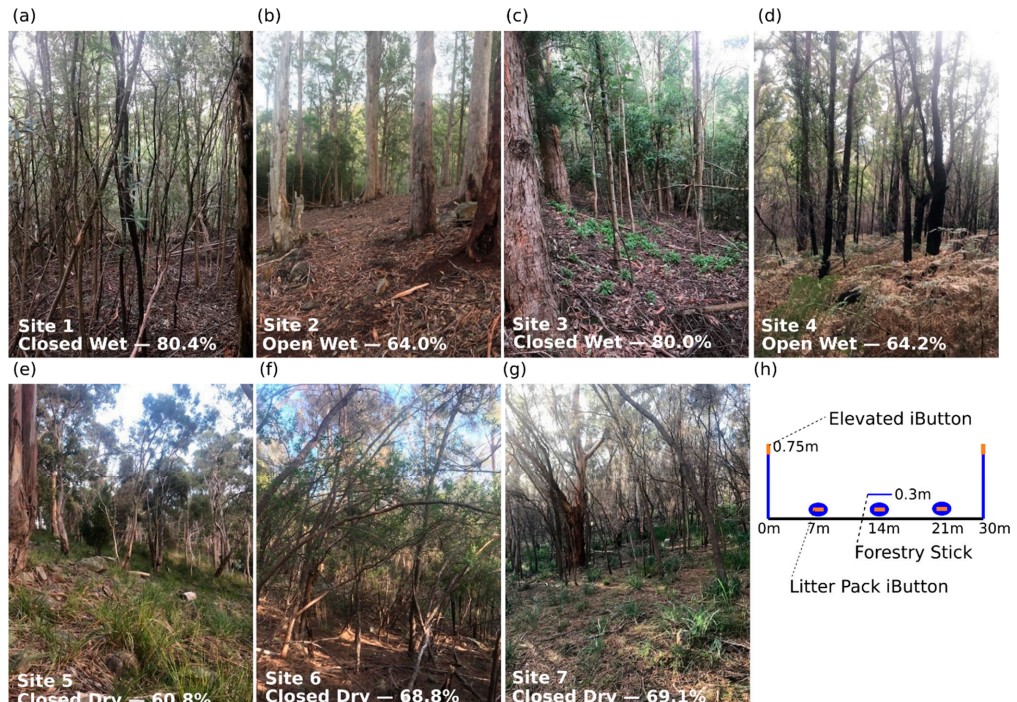

**Figure 2.** Photographs of the seven study sites (four wet and three dry *Eucalyptus* forests) near Hobart, Tasmania. Site number and site code, vegetation type, and fuel treatment are as follows: (**a**) Site 1 (PDC2)—wet forest that has remained unburned since 1967; (**b**) Site 2 (PDFB2)—wet forest subject to mechanical thinning removal of elevated fuels; (**c**) Site 3 (SAC4)—wet forest that has remained unburned since 1967 (**d**). Site 4 (SAPB4)—wet forest burned by a prescribed fire in March 2021; (**e**) Site

5 (LGMT3.NT)—dry forest unburned since 1997, subject to mechanical thinning; (**f**) Site 6 (LG PB1)—dry forest burned in 1997 and then by a prescribed fire in April 2021; (**g**) Site 7 (LGMT3.C)—dry forest unburned since 1997. Estimates of canopy cover percentage using a densitometer are indicated for each site is indicated. All the dry forest sites have northly aspects, whereas wet forest sites 1 and 2 had southerly aspects and sites 3 and 4 northerly aspects, albeit on the edge of a riparian zone. The forest types differ with respect to elevation and rainfall (wet 300–450 m vs. dry 250–300 m and wet 1000 mm vs. dry 600 mm per annum). (**h**) The layout on the transects of the suspended Hygrochron iButton, fuel moisture disks, and fuel moisture sticks is also shown.

## 2. Methods

### 2.1. Field Sampling

We monitored litter fuel moisture in wet and dry *Eucalyptus* forests with contrasting fuel management treatments and, hence, canopy covers (Figure 2a–g). We chose 4 locations in wet *Eucalyptus* forests (2 sites with and 2 sites without fuel treatment) and 3 locations in *Eucalyptus* dry forests (2 sites with and 1 site without fuel treatments) on the wildland–urban interface of Hobart, Tasmania, over the austral summer of 2021–2022 (Figures 1 and 2). Sampling occurred from 19 November 2021 until 4 March 2022, on an approximately weekly basis (Figure 3).

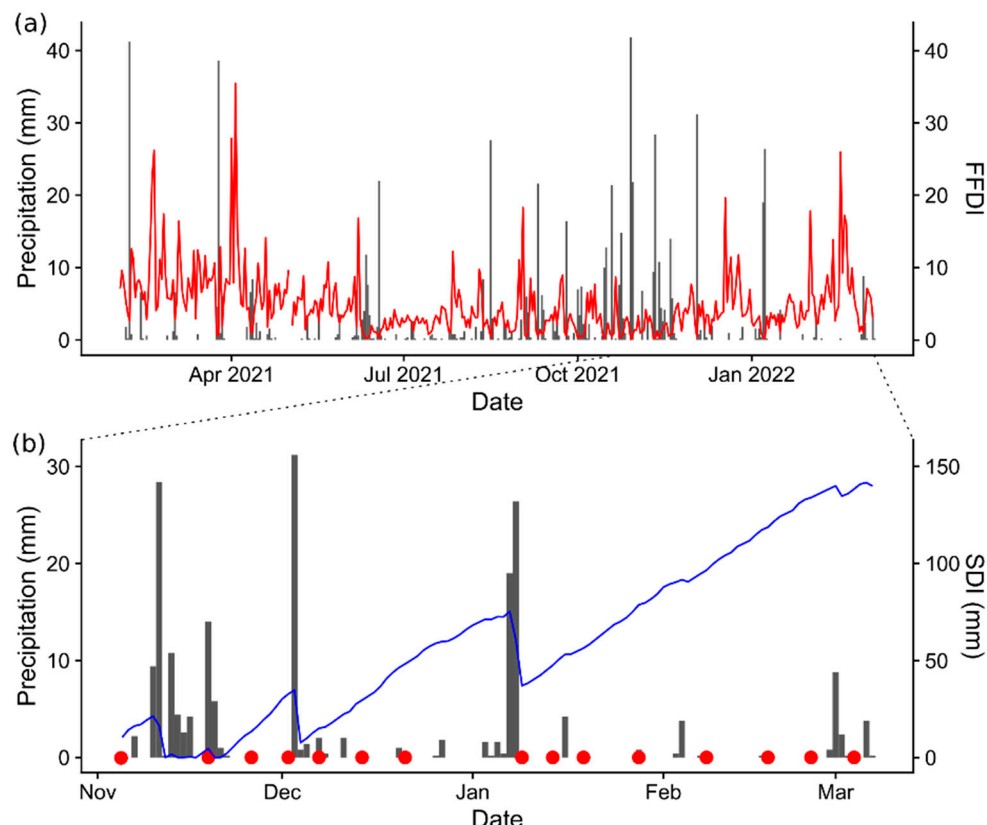

**Figure 3.** *Cont.*

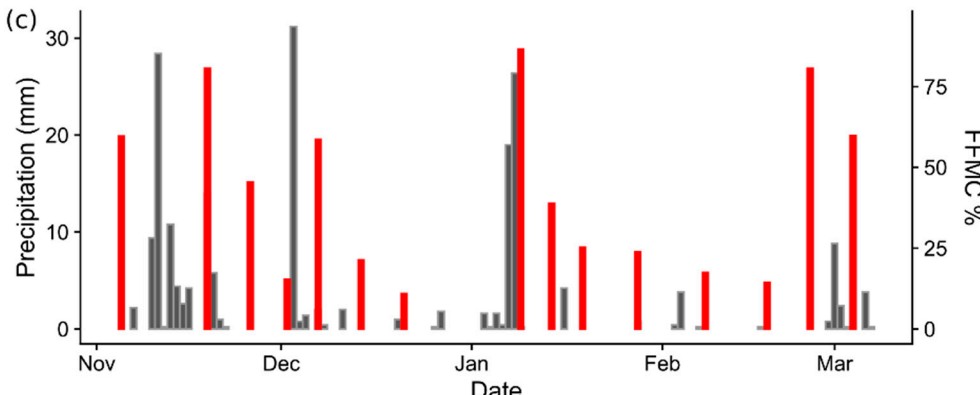

**Figure 3.** Precipitation and forest fire danger index prior to and during the 2021–2022 summer field campaign at Hobart, Tasmania. (**a**) Rain over preceding year (grey bars) and Forest Fire Danger Index (FFDI) (red line) based on data from nearest meteorological station (Hobart) for 12 months including 3-month sampling period. (**b**) Rainfall (grey bars) and Soil Dryness Index (SDI) (blue line) [20] during summer sampling campaign (defined by the dashed lines on the time axis of panel (**a**), where red dots indicate sampling dates. (**c**) Rainfall (grey bars) and mean gravimetric fine fuel moisture content percentage (FFMC) (red bars) on measurement dates during summer sampling campaign.

At each site, 30 m long monitoring transects were established (Figure 2h). Canopy cover was assessed using a densitometer held at 0.75 m above the ground surface and taken every 5 m on the transect: these seven measurements were averaged (Figure 2). Dead fuel moisture was measured using circular cages (litter disks) made from 40 mm ring of 225 mm diameter PVC tubing wrapped in 13 mm steel mesh (Figure 4a). The litter disks were spaced 7 m apart (7, 14, 21 m) on each transect and pinned to the ground. These were filled with leaf litter collected from the site, such that the litter mass matched the field density. The enclosed litter samples were allowed to equilibrate with the forest litter layer for at least one week and then collected between 11 am and 3 pm. They were taken to the laboratory in a sealed bag, making a note of the collection time. At the time of sampling a fresh litter disk was place in the field. In the lab, the litter mass was weighed, oven dried at 105 °C for at least 24 h, and reweighed to determine gravimetric moisture content. In each litter disk was an Hygrochron iButton, which measured humidity and temperature every hour, contained in the specially designed housing used by Nyman et al. [10]. To provide a proxy of fuel moisture routinely used by Tasmanian forest fire managers [22], an array of three *Pinus radiata* fuel moisture sticks, held together by plastic mesh, separated c. 2 cm apart, and supported by a wire frame 30 cm above the ground surface, was positioned in the middle of the transect (Figure 4b). At the same time as the litter sampling, the group of three fuel moisture sticks were weighed together in the field using an electronic balance with an accuracy of 0.01 g. At the beginning and end of the transect, Hygrochron iButtons were suspended 0.75 m above ground held in a specially designed fob and shielded from direct sunlight with a plastic cap (Figure 4c).

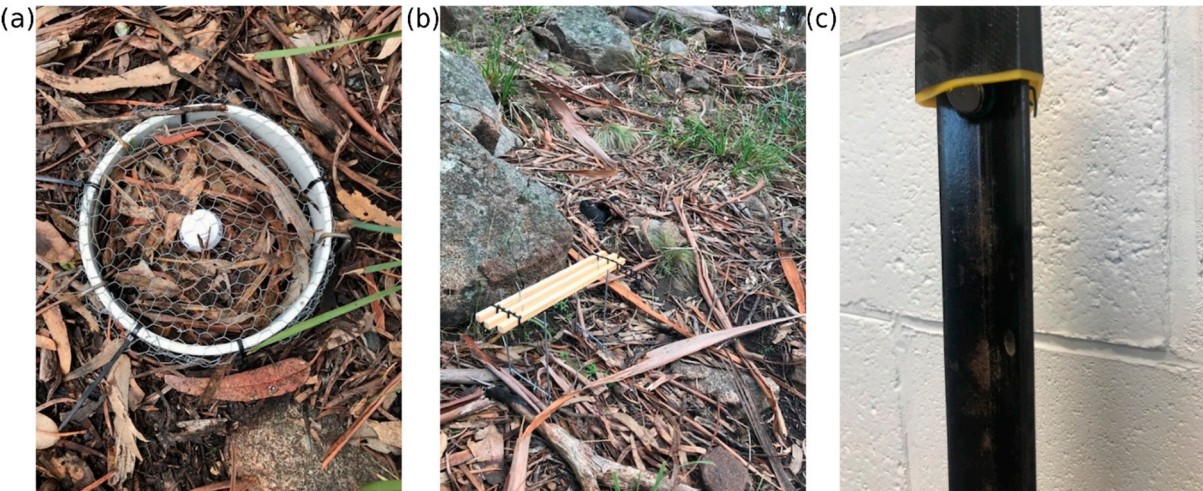

**Figure 4.** Photographs of equipment used to measure fine fuel moisture content. (**a**) Litter samples contained within circular cages that included a Hygrochron iButton held in a specially designed plastic case. (**b**) Three *Pinus radiata* fuel moisture sticks held together with plastic mesh that separates them c. 2 cm apart and supported by a wire frame 30 cm above the ground surface. (**c**) Hygrochron iButton attached to a steel stake using a specially designed fob and shielded from direct sunlight with a plastic cap.

*2.2. Data Analysis*

We used linear mixed models to evaluate the relationship of both gravimetric litter moisture content and fuel moisture stick moisture content with Mount's Soil Dryness Index (SDI) [20] calculated from the nearest meteorological station, with site included as a random effect to control for the repeated application of a single meteorological station to multiple sites, and forest type as a covariate. A one-way ANOVA and Tukey's multiple range tests were used to contrast the mean gravimetric FFMC across the summer measured at the seven sites. A linear mixed effect model with site as a random effect was used to establish the relationship between the gravimetric FFMC averaged across each transect with gravimetric moisture fuel moisture stick measurement made from the centre of the transect.

A suite of models was designed to predict gravimetric FFMC using Hygrochron iButton measurements of fuel moisture index (FMI), with 1 h and 24 h averaging, taken from within litter or elevated 75 cm above the litter with site as a random effect. Gravimetric FFMC was log-transformed based on diagnostic correlation plots that indicated a non-linear relationship with iButton measurements. Analysis was performed in R version 4.2.0 [23] using the packages lme4 1.1-29 [24] MuMIn 1.46.0 [25] and ggeffects v1.1.2 [26]. We used marginal effects plots from the ggeffects package in R [26] and marginal r2 calculations from the MuMIn package [25] to analyse the correlation in the model results.

**3. Results and Discussion**

With a few exceptions, the daily Forest Fire Danger Index (FFDI) was below 15, reflecting the cool conditions associated with a *la Nina* climate mode [27] (Figure 2a). There was high rainfall in the spring prior to the near continuously dry summer field sampling (Figure 2b,c). Despite two short-lived drops following significant rainfall (>30 mm), the Soil Dryness Index (SDI) from a nearby meteorological station steadily increased throughout the summer (Figure 2b). We found estimates of the SDI were poorly related to log gravimetric litter fuel moisture and gravimetric fuel moisture stick moisture content (marginal $r^2$ = 8.2% and 3.7%, respectively, Supplementary Figure S1), reflecting strongly different effects of rain on soil recharge and FFMC (Figure 2b,c). There were significant differences in gravimetric FFMC amongst the seven sites (Figure 5). Broadly, the wet forests had higher and dry forests lower FFMC. It is important to note that wet forest sites 1 and 2 were on a polar-facing

slope, while wet forest sites 3 and 4 were on an equatorial facing slope, which explains the differences between them, given the strongly contrasting effects aspect at this latitude (42.8° S) [28]. The effect of recent disturbance on FFMC in dry forests was negligible (Figure 5). In wet forests, recent disturbance (both mechanical treatment and prescribed burning) significantly reduced FFMC, making it comparable with FFMC in dry forests (Figure 5). The greater effect of disturbance in wet forests than dry forests on FFMC most probably reflects the importance of understorey vegetation type in controlling the microclimate in wet forests [19,29].

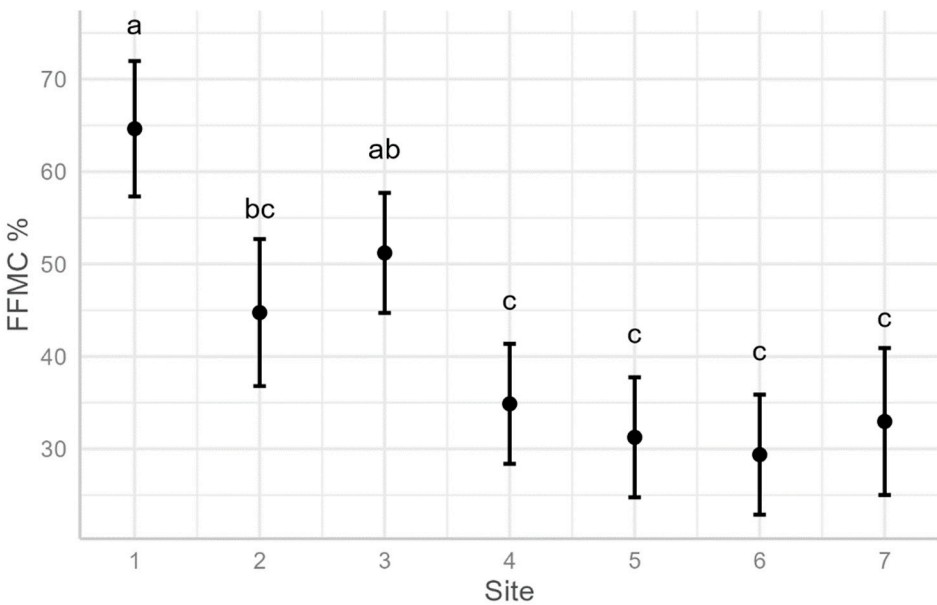

**Figure 5.** Mean and 95% confidence intervals of the gravimetric fine fuel moisture content (FFMC) in four wet and three dry *Eucalyptus* forest sites surrounding Hobart, Tasmania (Figures 1 and 2). Letters denote means that are statistically similar according to Tukey's multiple range tests, where significance differences are defined as *p* < 0.05).

A linear regression of gravimetric litter fuel moisture content against fuel moisture stick measurements showed a strong association (marginal $r^2$ = 67.5% deviance, Figure 6a), particularly at low FFMC, confirming that both measures are representing site FFMC. Modelling of log-transformed gravimetric fuel moisture content against the various FMI measurement, with site as a random effect, showed the strongest relationship was with 24 h average FMI, measured by an Hygrochron iButton 0.75 m above the surface (marginal $r^2$ = 58.8%, Figure 6b) (Table 1). Site had a negligible statistical effect on the gravimetric FFMC and 24 h average FMI.

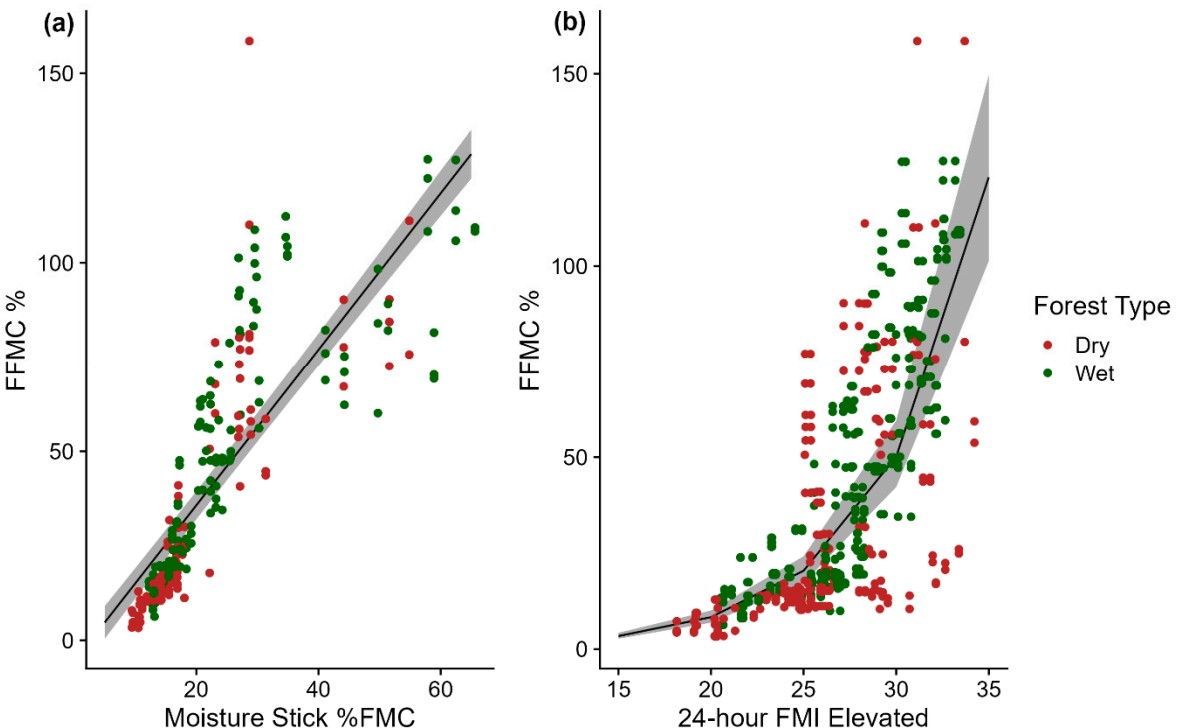

**Figure 6.** Plots of percentage of litter fine fuel moisture content, percentage of gravimetric fuel moisture stick fuel moisture content, and 24 h Fuel Moisture Index (FMI). (**a**) Percentage of gravimetric fuel moisture stick (c. 10 h fuels) moisture content (% FMC) plotted against percentage of gravimetric litter (c. 1 h fuels) fine fuel moisture content (FFMC %). (**b**) Highest-ranking linear mixed model that relates 24 h 0.75 m elevated FMI against log gravimetric litter fine fuel moisture content (FFMC %). Forest types are indicated by coloured dots, where red = dry forests and green = wet forests.

**Table 1.** Akaike information criterion (AIC) statistics for linear mixed models, with site as a random effect for log fuel moisture content predicted by FMI.

| Model | K | AICc | ΔAICc | AICc Weight | Log Lik. | Cum. Weight | $Mr^2$ | RMSE |
|---|---|---|---|---|---|---|---|---|
| FMI 24 h Elevated | 4 | 761.5 | 0.00 | 1 | −376.7 | 1 | 0.588 | 0.49 |
| FMI 1 h Elevated | 4 | 923.2 | 161.6 | 0 | −457.5 | 1 | 0.531 | 0.58 |
| FMI 1 h Litter Pack | 4 | 992.0 | 230.4 | 0 | −492.0 | 1 | 0.354 | 0.61 |
| FMI 24 h Litter Pack | 4 | 1124.0 | 362.5 | 0 | −558.0 | 1 | 0.221 | 0.70 |
| Null | 3 | 1229.5 | 467.9 | 0 | −611.7 | 1 | 0.000 | 0.77 |

Based on the linear mixed model and using the mean intercept across sites, the relationship between FMI and gravimetric FFMC in this system can be defined by Equation (2).

$$\text{FFMC} = e^{\text{i}+\text{b FMI}} \tag{2}$$

where intercept i = −1.474 (s.e. = 0.19) and slope b = 0.1796 (s.e. = 0.0064).

Like previous fine fuel moisture studies in *Eucalyptus* forests, we found a strong effect of wet and dry *Eucalyptus* forest type [8,10,29], time since disturbance [7] (Figure 4), and understorey cover [7,29] (Figure 5). Our study also aligns with other studies that have shown that FMI is a simple and effective index of fuel moisture in *Eucalyptus* forests during dry conditions [8,17,30], noting that here we have used FMI as a dimensionless index rather than adjusting using a scaling factor, which assumes a linear relationship between FFMC and FMI, as was applied in those previous studies. Given the strong linear correlation between gravimetric FFMC and the fuel moisture sticks (Figure 6a), our FMI estimate, which was based on 24 h surface temperature and humidity estimates, is predictive of both

1 h and 10 h fine fuel moisture (Figure 3a). Concordant with the analysis of Nyman et al. [10], we show that the relationship of elevated FMI with gravimetric FFMC is curvilinear, with increasing scatter at higher moisture contents, aligning with previous findings that FMI performs better in drier fuels [8,17]. In contrast to Nyman et al. [10], however, we found that Hygrochron iButtons embedded in litter did not produce estimates of FMI that were highly predictive of FFMC.

The flammability of Australian forest fine fuels has received limited attention [31,32], particularly the specific FFMC threshold that switches fuels to flammable [5], although [33] suggests *Eucalyptus* litter becomes flammable at 16% moisture content. Nolan et al. [34] undertook a landscape ecology analysis of the relationship between fuel moisture and wildfire occurrence and extent. These authors found that a modelled dead fuel moisture content (FM) of c. 30% controlled occurrence, with a dead FM threshold of c. 15%, 10%, and 5% relating to step changes in increasing fire size. Based on mean mixed-model intercepts (Table 1), our analysis suggests FMI values of 27, 23, 21, and 17 would equate to FFMC contents of 30%, 15%, 10%, and 5%, respectively, for the Tasmanian wet and dry forest types we studied.

In summary, this study endorses the use of a 24 h average Fuel Moisture Index, based on Hygrochron iButton data humidity temperature loggers suspended 0.75 m above the forest floor, to estimate fine fuel moisture content (FFMC) in Tasmanian wet and dry *Eucalyptus* forests subjected to strongly contrasting disturbance histories. This approach has the potential to further advance the understanding of the relative effect of vegetation type and disturbance history in shaping landscape flammability, fire regimes, and vegetation patterns in Australia [5,35] and elsewhere in the world [36].

**Supplementary Materials:** The following supporting information can be downloaded at: https://www.mdpi.com/article/10.3390/fire5050130/s1. Figure S1: Effect plots of log fuel moisture content and fuel moisture stick measurements against Soil Dryness Index (SDI).

**Author Contributions:** Conceptualization, D.M.J.S.B.; methodology, D.M.J.S.B., J.M.F. and M.P.; formal analysis, G.J.W.; writing—original draft preparation, D.M.J.S.B.; writing—review and editing, J.M.F., M.P. and G.J.W.; project administration, M.P.; funding acquisition, D.M.J.S.B. All authors have read and agreed to the published version of the manuscript.

**Funding:** This research was funded by Australian Research Council grant DP200102395 and Tasmanian State Emergency Service Natural Disaster Risk Reduction Grants Program project B0028205 'The Social and Biophysical Effects of Alternative Strategies to Reduce Bushfire Danger in Hobart'.

**Institutional Review Board Statement:** Not applicable.

**Informed Consent Statement:** Not applicable.

**Data Availability Statement:** Soil and fuel moisture datasets are available in a repository at https://doi.org/10.6084/m9.figshare.20305317.

**Acknowledgments:** We thank Jane Cawson and Petter Nyman for providing the designs for the iButton casing and advice on the methodology, and Paul Fox-Hughes, Australian Bureau of Meteorology, and Dean Williams, Sustainable Timbers Tasmania, for their helpful comments on this manuscript.

**Conflicts of Interest:** The authors declare no conflict of interest.

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
