# Peer review of "The Fuel Moisture Index Based on Understorey Hygrochron iButton Humidity and Temperature Measurements Reliably Predicts Fine Fuel Moisture Content in Tasmanian Eucalyptus Forests"

_fire, doi:10.3390/fire5050130_

Round 1
Reviewer 1 Report
Dear authors,
I forward my suggestions for minor changes to improve this technical note.

Reviewer 2 Report
This manuscript uses the 24-hour average fuel moisture index to estimate fine fuel moisture content (FFMC) in wet and dry eucalyptus forests in Tasmania. This research is based on a very detailed field survey. This study was clear and informative and provided further insight into the relative influence of vegetation type and disturbance history in shaping landscape flammability, fire intensity, and vegetation patterns. I recommend minor revisions prior to publication. Some comments that might improve this work are listed below.
1) Page 1, Line 15 and 19, there is a spelling error. “gravimentric” or “gravimetric”.
2) Page 1, Line 28, What is the meaning of the "c." that appears many times before the number in the article?
3) Page 2, Line 53, The symbol for degrees Celsius is incorrect. (℃)
4) Figure 3, Add a legend to make the graph clearer. Does the dashed line in Figure 3(b) make sense?
5) The authors show the relationship between precipitation and FFDI or SDI, what about the relationship between precipitation and FFMC?
6) What is the relationship between SDI and fire risk? It is recommended to cite relevant literature for support.
7) Page 7, Line 171-177, Is it surprising that recent disturbances have affected wet and dry forests so differently?
8) Can a more detailed explanation of the differences of FFMC between the seven different sites be provided?
9) Figure 5 and Figure 6, The labeling of the ordinate should be “gravimetric FFMC (%)”.
10) Hopefully the article will give a more detailed explanation of the "strongly contrasting interference history".
11) The research area in the article is the Tasmanian forest in Australia, so can this research be applied to larger-scale areas or different vegetation types?
12) Several references below may be considered in this work.
Cruz, M. G., S. Kidnie, S. Matthews, R. J. Hurley, A. Slijepcevic, D. Nichols, and J. S. Gould, 2016: Evaluation of the predictive capacity of dead fuel moisture models for Eastern Australia grasslands. International Journal of Wildland Fire, 25, 995.
Nolan, R. H., V. Resco de Dios, M. M. Boer, G. Caccamo, M. L. Goulden, and R. A. Bradstock, 2016: Predicting dead fine fuel moisture at regional scales using vapour pressure deficit from MODIS and gridded weather data. Remote Sensing of Environment, 174, 100-108.
Quan, X., M. Yebra, D. Riaño, B. He, G. Lai, and X. Liu, 2021: Global fuel moisture content mapping from MODIS. Int J Appl Earth Obs, 101.
Slijepcevic, A., W. Anderson, S. Matthews, and D. Anderson, 2015: Evaluating models to predict daily fine fuel moisture content in eucalypt forest. Forest Ecology and Management, 335, 261-269.
Reviewer 3 Report
The technical note addresses the role of using Hygrochron iButton humidity and temperature measurements to determine the Fuel Moisture Index in Tasmanian wet and dry Eucalyptus forests. The results are compared with the gravimetric fuel moisture content which provides real/ observed data but depends on field sampling. The work achieved interesting findings for the research area. I consider that the approach followed is simple and effective to predict the fuel moisture content, and consequently, the daily fire danger. I recommend the technical note for publication after minor revisions. In the attached PDF file the suggested minor revisions are presented.
